# Rejection Sensitivity and Problematic Internet Use Among Medical Students: A Moderated Mediation Model Involving Loneliness and Self-Control

**DOI:** 10.3390/bs15050589

**Published:** 2025-04-27

**Authors:** Cheng Xu, Meiling Liao, Youjuan Hong

**Affiliations:** 1School of Nursing, Fujian Medical University, Fuzhou 350122, China; xvcheng0830@126.com; 2School of Health, Fujian Medical University, Fuzhou 350122, China; 3Research Center for Nursing Humanity, Fujian Medical University, Fuzhou 350122, China

**Keywords:** rejection sensitivity, loneliness, self-control, problematic internet use

## Abstract

The internet has evolved into an integral facet of the life and educational experience of college students. However, the driving force behind problematic internet use in medical students remains largely unexplored. Based on the cognitive–behavioral model and the affect regulation model of vulnerability, the present study seeks to investigate the underlying mechanism of the association between rejection sensitivity and problematic internet use among medical students. A total of 435 undergraduate medical students were investigated using the Tendency to Expect Rejection Scale, Loneliness Scale, Self-control Scale, and Problematic Internet Test. The results revealed that rejection sensitivity could positively predict loneliness and problematic internet use. Mediation analyses revealed that loneliness served as a mediator in the link between rejection sensitivity and problematic internet use. Additionally, self-control moderated the second stage of the indirect effects in the relationship between rejection sensitivity and problematic internet use. These findings theoretically deepen our understanding of the psychological pathways and the boundary conditions linking medical students’ rejection sensitivity to their problematic internet use, while also offering valuable practical implications for decreasing problematic internet use.

## 1. Introduction

The utilization of the internet has witnessed a substantial surge in the last two decades ([8]). Studies indicate that 73% of college students engage with the internet daily, dedicating approximately 1.6 to 4.5 h online ([12]). Prolonged engagement in online activities can contribute to problematic internet use, characterized by risky, excessive, or impulsive internet behavior, leading to detrimental consequences in various life domains, including physical, emotional, social, or functional impairment ([28]). The prevalence of problematic internet use (PIU) in the college-based population was reported to be around 9.22% ([36]). Medical students are a highly academically stressed group who are vulnerable to psychopathologies, and therefore they are at risk of adopting escapist coping styles such as PIU ([2]). Existing studies indicate a PIU detection rate among Chinese medical students of approximately 30.1%, significantly surpassing that observed in non-medical undergraduate students ([50]). The PIU among medical students could detrimentally affect their academic performance and contribute to an increased prevalence of insomnia ([33]; [4]) and psychiatric disorders such as depression, anxiety, social withdrawal, and attention deficit ([19]). This situation can have profound repercussions for medical students with aspirations to become healthcare professionals, disrupting their academic pursuits and jeopardizing their long-term career objectives ([34]). Additionally, it may entail extensive and adverse implications for the future physicians’ capacity to deliver high-quality care to patients and the broader community ([14]).

Previous studies focused on the prevalence and negative effect of PIU among medical students ([3]; [13]; [26]; [38]), while few studies have investigated the driving factors and inner mechanism (such as personality traits, emotional factors) of PIU among medical students. Given the scarcity of empirical studies in this domain and the escalating prevalence of internet usage in China, particularly among medical students, it is imperative to enhance comprehension of the underlying factors and associated mechanisms of PIU. This knowledge is crucial for preventive efforts to prevent medical students from succumbing to pathological internet use. Individual personality traits (e.g., rejection sensitivity), emotional states (e.g., loneliness), and individual positive qualities (e.g., self-control) may be closely associated with PIU in medical students. However, research has not yet explored how and when these factors influence the PIU of medical students ([7]; [46]). Therefore, this study aims to investigate the mechanistic impact of individual risk factors and positive factors on PIU among medical students.

### 1.1. Rejection Sensitivity and PIU

Rejection sensitivity refers to the tendency to anxiously expect and readily perceive rejection from others ([40]). These individuals tend to readily construe ambiguous interpersonal situations as instances of social rejection, projecting negative responses to actual or perceived emotions ([37]). Research confirmed that individuals with a trait of fearing rejection are more likely to engage in pathological internet use behaviors, highlighting rejection sensitivity as a threatening factor triggering pathological internet use ([11]). Individuals with high rejection sensitivity, after experiencing social rejection, tend to immerse themselves in the online realm to seek a sense of security ([29]). Medical students with elevated rejection sensitivity may find greater comfort in online interactions as opposed to face-to-face interactions, attributed to their heightened sensitivity to interpersonal rejection ([42]). Therefore, rejection sensitivity may increase medical students’ PIU. Therefore, we posit the following:
**Hypothesis** **1.***There is a positive correlation between rejection sensitivity and pathological internet use among medical students.*

### 1.2. Loneliness as a Mediator

Loneliness is characterized as an individual’s adverse emotional reaction to social isolation and absence of companionship, frequently accompanied by feelings of anxiety ([35]). Previous study found that individuals with high rejection sensitivity often generate negative emotions such as hostility in social interactions and tend to adopt avoidance coping strategies, negatively impacting the establishment and maintenance of positive interpersonal relationships ([20]), leading to the experience of loneliness ([43]). Therefore, individuals with elevated rejection sensitivity demonstrate increased levels of loneliness ([9]). According to Davis’s cognitive–behavioral model ([6]), individuals with maladaptive cognitions may perceive the online world as more capable of fulfilling their needs compared to real-life experiences. Individuals experiencing loneliness tend to favor online interaction over face-to-face interaction due to perceived deficits in social skills. They may use online interaction as a compensatory mechanism, akin to a form of social ‘Prozac’, leading to compulsive internet use and subsequent adverse outcomes, such as PIU. Numerous studies confirmed that individuals with higher emotional experiences of loneliness are more prone to developing tendencies toward internet addiction ([39]; [1]; [32]; [25]). Prasanna et al.’s study identified a positive correlation between loneliness and internet use in medical students ([31]). Therefore, the rejection sensitivity of medical students may influence their problematic internet use through the mediating role of loneliness. Therefore, we posit the following:
**Hypothesis** **2.***Loneliness mediates the impact of rejection sensitivity on problematic internet use among medical students.*

### 1.3. Self-Control as a Moderator

Self-control, as defined by Duckworth ([10]), encompasses the ability to monitor, inhibit, persevere, and adapt one’s behavior, emotions, thoughts, and desires to achieve specific goals. Individuals with robust self-control demonstrate superior proficiency in regulating behavioral, emotional, and attentional impulses to attain long-term objectives ([31]). [48] ([48]) posited that PIU is essentially an impulse control issue ([48]). Previous research has indicated that self-control plays a significant role in PIU, surpassing the impact of other variables ([45]). Higher levels of self-control are associated with a decreased likelihood of internet addiction. According to the affect regulation model of vulnerability, individuals facing challenges in emotional control are predisposed to addiction problems, with the intensity of this correlation contingent upon their level of self-control ([47]). For medical students with low self-control, individuals with high loneliness tend to excessively rely on the internet as a tool for interpersonal communication, making it challenging to control their internet use, consequently leading to PIU. [21] ([21]) explored the moderating influence of self-control in the relationship between school connectedness, deviant peer affiliation, and PIU ([21]). Thus, self-control may moderate the relationship of loneliness and PIU. Therefore, we posit the following:
**Hypothesis** **3.***Self-control moderates the influence of loneliness on problematic internet use among medical students.*

### 1.4. The Hypothesized Model of Study

To sum up, the study constructs a moderated mediation model (Figure 1) to investigate the psychological pathway and boundary conditions of rejection sensitivity on PIU among medical students. Specifically, this study examines the mediating role of loneliness in the association between rejection sensitivity and PIU and the moderating role of self-control in the mediation model.

## 2. Method

### 2.1. Participants and Procedure

This study employed a convenience sampling method, enlisting 486 medical students from Fuzhou City, Fujian Province. After excluding 51 incomplete or invalid questionnaires, 435 valid questionnaires were collected, resulting in a response rate of 89.50%. The sample comprised 217 male students and 218 female students, with 136 first-year students, 167 second-year students, 56 third-year students, 55 fourth-year students, and 21 fifth-year students. Furthermore, there were 140 only children and 295 non-only children, 213 urban students, and 222 rural students, with an average age of (20.26 ± 4.74) years. The inclusion criteria were female and male medical students studying in medical universities and the absence of severe psychotic disorders and substance use.

The study obtained ethical approval from the Institutional Review Board (IRB) of Fujian Medical University. Trained research assistants distributed the questionnaires, introducing the survey’s purpose and completion requirements to the participants. After obtaining informed consent, the assistants distributed and collected the questionnaires. The study emphasized the authenticity, independence, and anonymity of all responses provided by the participants. Students were informed about the voluntary nature of their participation and their right to terminate their involvement at any time. Participants received a token of appreciation as incentives.

### 2.2. Measurements

#### 2.2.1. Rejection Sensitivity

The Tendency to Expect Rejection Scale was utilized to assess rejection sensitivity, a measure previously validated for Chinese young adults ([17]; [49]). The Chinese version used in this study demonstrated acceptable reliability and validity ([29]; [22]). Participants rated 18 items on a 5-point scale (ranging from 1 to 5) to indicate the extent to which certain characteristics applied to them. Sample items include “I’m sensitive to rejection” and “It’s important to me to be accepted by those around me”. The mean score was calculated. A higher total score indicates elevated levels of rejection sensitivity. The Cronbach’s alpha in the current study was 0.82.

#### 2.2.2. Loneliness

The Loneliness Scale, a modified version by Hays and DiMatteo ([16]) derived from the UCLA-20 Loneliness Scale, comprises 8 items, including statements like “I feel isolated from others” and “There is no one I can turn to”. Participants rated each item on a Likert scale ranging from 1 (Never) to 4 (Always). A higher overall score indicates an elevated level of loneliness. In the present study, the Cronbach’s alpha was 0.81.

#### 2.2.3. Self-Control

Self-control was measured using the Brief Self-Control Scale (BSCS), developed by [27] ([27]) and revised by [23] ([23]). The BSCS comprises 7 items assessing two dimensions: self-discipline and impulse control ([51]). Self-discipline assesses an inclination toward deliberate/non-impulsive action and includes 3 items such as “I am good at resisting temptation”, while impulse control was negatively worded and included 4 items such as “I do certain things that are bad for me, if they are fun” and “I can’t stop myself from doing something, even if I know it’s wrong”. Respondents used a 5-point Likert scale ranging from “Not at all like me” to “Exactly like me”. The impulse control items are typically calculated using reverse scoring. Higher scores on the scale reflect a greater level of self-control. In the current study, the Cronbach’s alpha for the scale was 0.76.

#### 2.2.4. Problematic Internet Use

The evaluation of problematic internet use employed the Internet Addiction Test (YIAT) developed by [48] ([48]), which consists of 20 items such as “How often does your job performance or productivity suffer because of the Internet?”. Each item is rated on a scale from 1 to 5, where 1 represents “not at all” and 5 represents “always”. Higher scores indicating more significant problems associated with internet usage. The scale categorizes participants into three categories: average users (≤49), maladaptive users (50–79), and pathological users (≥80). In the present study, the Cronbach’s alpha was 0.89.

### 2.3. Data Analysis

Firstly, the study calculated descriptive statistics and bivariate correlations for the variables. Secondly, the research adhered to [24] ([24]) four-step procedure for assessing the mediation effect. Thirdly, the study explored the potential moderation of the mediation process by self-control. Moderated mediation is frequently utilized to assess whether the extent of a mediation is contingent on the value of a moderator ([15]). The analysis of the moderated mediation model was executed using the PROCESS macro (Model 14). To determine the 95% confidence intervals (CIs) of bias-corrected bootstrapping, the number of bootstrap samples used in the present study was 5000.

## 3. Results

All data in this study were obtained through subjective self-report measures, and the results may be influenced by common method bias. Therefore, this study conducted Harman’s single-factor test. The findings revealed 13 factors with eigenvalues exceeding 1 in the unrotated solution, elucidating 58.63% of the variance. The initial factor accounted for 18.65% of the variance, falling below the critical threshold of 40%. Consequently, the impact of common method bias on the study is deemed acceptable.

### 3.1. Preliminary Analyses

Table 1 presents the means, standard deviations, and correlations among the variables employed in each analysis. As expected, rejection sensitivity has a weak positive correlation with loneliness (*r* = 0.39) and pathological internet use (*r* = 0.36) and is significantly negatively correlated with self-control (*r* = −0.21). Loneliness is significantly negatively correlated with self-control (*r* = −0.24) and significantly positively correlated with pathological internet use (*r* = 0.28). Self-control is moderate negatively correlated with pathological internet use (*r* = −0.55). Hypothesis 1 is supported.

### 3.2. The Mediation Effect of Loneliness

Firstly, the study utilized Hayes’s PROCESS macro (Model 4, 2013) to examine the mediating role of loneliness in the association between rejection sensitivity and PIU. The analyses incorporated gender, age, family location, and grade as covariates. The initial step of multiple regression analysis revealed a significant positive relationship between rejection sensitivity and PIU, with *b* = 0.39, *p* < 0.001 (see Table 2). In the second step, rejection sensitivity was significantly associated with loneliness, *b* = 0.33, *p* < 0.001. In the third step, when this model controlled for rejection sensitivity, loneliness was significantly related to PIU, *b* = 0.27, *p* < 0.05. Finally, the bias-corrected percentile bootstrap method showed that the indirect effect of rejection sensitivity on PIU via loneliness was significant, ab = 0.06, *SE* = 0.02, 95%CI = [0.02, 0.10]. The mediating effect explained 18.18% of the total effect (see Table 3). All four criteria for establishing a mediation effect were met. Thus, Hypothesis 2 was supported.

### 3.3. The Moderation Effect of Self-Control

This study utilized Macro Process Model 14 to examine the moderating effect of self-control on the relationship between loneliness and PIU. The interaction effect between self-control and loneliness significantly and positively predicted PIU (*b* = 0.08, *SE* = 0.04, *t* = 2.35, *p* < 0.05), as indicated in Table 4. To provide additional insights into this interaction, self-control was stratified into high and low groups using one standard deviation above and below the mean, and simple slope tests were executed. As depicted in Figure 2, for individuals with low self-control, loneliness had a stronger positive impact on PIU (*b* = 0.06, *SE* = 0.02, *p* < 0.001); for individuals with high self-control, the predictive effect of loneliness on PIU was smaller (*b* = 0.02, *SE* = 0.01, *p* < 0.05). Therefore, Hypothesis 3 was supported (see Table 5).

## 4. Discussion

This study, grounded in the cognitive–behavioral model and the affect regulation model of vulnerability, investigates the intrinsic motivations and influencing mechanisms of PIU among medical students, considering individual personality traits, emotional states, and other risk factors, as well as positive factors such as abilities. Building upon existing research ([2]; [5]), it further explores the intrinsic impact of interpersonal factors on PIU among medical students. The findings disclose a positive correlation between rejection sensitivity and PIU among medical students. Additionally, the mediation effect of loneliness in the relationship between rejection sensitivity and PIU is moderated by the self-control of medical students. The study findings, employing an individual factors perspective and utilizing empirical methods, substantiated both the risk and protective factors linked to problematic internet use among medical students, offering valuable insights for the prevention of such issues in this demographic.

This study found that rejection sensitivity positively predicts PIU in medical students, which contributes to highlighting the connection between rejection sensitivity and PIU. This conclusion aligns with previous research ([11]; [42]). Medical students with high rejection sensitivity may experience unmet needs for belongingness and perceive weaker social support, leading them to engage excessively in online activities as a compensatory mechanism for the lack of a sense of belonging and to fulfill their emotional support needs ([44]). Specifically, problematic internet use may function as a maladaptive coping strategy aimed at regulating stressful emotions ([11]). Previous studies have focused more on the effects of academic stress or anxiety on medical students’ PIU ([50]) and have not paid attention to the effects of rejection sensitivity, which is related to interpersonal interactions, on medical students’ PIU. The findings confirm that rejection sensitivity is also a risk factor for problematic internet use among medical students.

This study also showed that loneliness mediates the relationship between rejection sensitivity and PIU among medical students, which is consistent with the cognitive–behavioral model ([6]). This model posits that PIU results from problematic cognitions coupled with behaviors that intensify the maladaptive response. Medical students with high rejection sensitivity not only tend to misinterpret ambiguous social cues and exhibit hypervigilance ([42]) but also struggle to express emotional needs and form positive social connections in reality. This dual deficit frequently culminates in loneliness that reflects both unsatisfied social needs and impeded socioemotional development ([43]; [41]). To compensate for these interpersonal deficiencies, they resort to seeking virtual solace through excessive online engagement, creating a maladaptive cycle of compensatory internet use ([18]). Our findings provide the first empirical study of loneliness as the vital mediating factor for the association between rejection sensitivity and medical students’ PIU. This study identified the pathway of rejection sensitivity to PIU among medical students and further clarified the risk drivers of PIU among medical students.

In this study, we also examined potential variations in the direct and/or indirect pathways based on the level of self-control among medical students. The findings indicate that self-control moderates the indirect relationship between loneliness and PIU among medical students. The finding aligns with the affect regulation model of vulnerability which indicates impaired emotional regulation heightens addiction susceptibility, moderated by self-control capacity ([47]). It is noteworthy that self-control “attenuates” the association between loneliness and PIU among medical students. This moderation effect implies that self-control may moderate the influence of risk factors (loneliness and PIU in this study) and function as a protective factor against PIU ([30]). To our knowledge, this study represents the initial investigation into the moderating influence of self-control among medical students in the relationship between loneliness and PIU.

## 5. Implication

Our findings have important practical implications. First, as rejection sensitivity positively correlated with PIU among medical students, educational administrators need to pay attention to medical students’ sensitivity in interpersonal relationships and its negative impact. Conducting relevant social skills courses and diverse group socialization activities, as well as creating an inclusive and tolerant microsystem environment, can increase social interactions among medical students and reduce their rejection sensitivity, preventing them from compensating through excessive use of the Internet. Second, our research findings provide insights for practitioners into the mediating role of loneliness in the influence of rejection sensitivity on PIU. The risk of PIU can be reduced by reducing rejection sensitivity and loneliness in medical students through counseling or social–emotional competence training or by developing rich and varied community activities. Third, self-control moderates the relationship between loneliness and PIU. Therefore, educators need to recognize the role of self-control in preventing PIU in medical students and implement self-control-enhancing curricula to improve medical students’ self-control.

## 6. Limitations and Future Study

This study has some limitations. Firstly, due to its cross-sectional design, it is challenging to establish definitive causal relationships between variables, and the potential impact of social desirability bias on self-reporting by medical students should be acknowledged. Future research could employ longitudinal designs and experimental studies to better collect data and further examine the inner mechanisms of rejection sensitivity on internet addiction. Secondly, as the participants in this study were limited to one city in mainland China, caution should be exercised when generalizing these results to other regions. Future research could expand the sampling area and size to explore whether the mediation model established in this study is universal, and medical students were not compared with other types of students. However, medical students can be compared with other types of students to emphasize the unique characteristics of medical students. Thirdly, in the current study, we assessed the PIU of medical students mainly in terms of duration, overdependence, degree of impact of internet access on academics, and annoyance at not being able to access the internet, without classifying specific types of internet use. In future studies, we could categorize internet use into specific subcategories, such as video games and chatting, and explore whether the effects of loneliness and self-control on the association between rejection sensitivity and PIU depend on a particular type of internet activity. Lastly, rejection sensitivity includes an emotional component and a cognitive component. However, this study only investigated the effect of overall rejection sensitivity on PIU, and it is possible that these two dimensions have different effects on PIU. Future studies could further investigate the effect of specific dimensions of rejection sensitivity on PIU.

## 7. Conclusions

Based on the cognitive–behavioral model and the affect regulation model of vulnerability, the study advances our empirical understanding as to how and when medical students’ rejection sensitivity influences their PIU. Our findings suggest that rejection sensitivity predicts PIU and loneliness operates as a potential mechanism mediating this relationship. Furthermore, moderated mediation findings indicate that self-control acts as a moderator in the association between loneliness and PIU among medical students.

## Figures and Tables

**Figure 1 behavsci-15-00589-f001:**
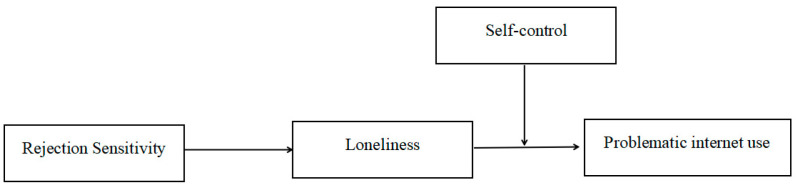
Hypothesized research model.

**Figure 2 behavsci-15-00589-f002:**
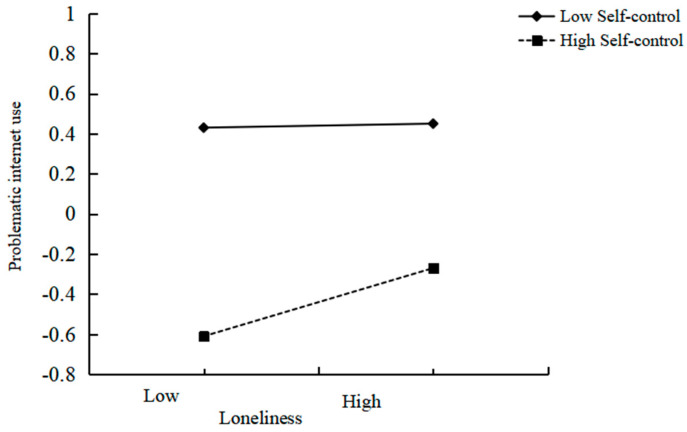
Plot of the relationship between loneliness and PIU at two levels of self-control.

**Table 1 behavsci-15-00589-t001:** Descriptive statistics and correlations of the main study variables.

	*M*	*SD*	1	2	3	4
1. Rejection Sensitivity	3.13	0.52	1			
2. Loneliness	2.40	0.38	0.39 ***	1		
3. Self-control	2.84	0.60	−0.21 ***	−0.24 ***	1	
4. PIU	2.86	0.57	0.36 ***	0.28 ***	−0.55 ***	1

*** *p* < 0.001.

**Table 2 behavsci-15-00589-t002:** Testing for mediation effect of rejection sensitivity on PIU.

Predictors		Model 1 (PIU)			Model 2 (Loneliness)			Model 3 (PIU)	
	*b*	*t*	Boot 95%CI	*b*	*t*	Boot 95%CI	*b*	*t*	Boot 95%CI
Gender	0.04	0.47	−0.14, 0.22	0.06	0.71	−0.10, 0.22	0.03	0.36	−0.14, 0.20
Age	0.01	0.28	−0.11, 0.11	−0.03	−0.63	−0.11, 0.06	0.01	0.20	−0.07, 0.11
Family location	0.09	1.14	−0.15, 0.11	0.11	1.38	−0.04, 0.26	0.04	0.57	−0.09, 0.24
Grade	−0.04	−0.57	−0.07, 0.28	0.01	0.21	−0.10, 0.12	−0.02	−0.54	−0.15, 0.08
Rejection sensitivity	0.39 ***	7.94		0.33 ***	8.38	0.25, 0.41	0.27 *	5.93	0.18, 0.36
Loneliness							0.18 ***	3.47	0.07, 0.27
*R* ^2^	0.13			0.15			0.15		
*F*	12.99 ***			15.77 ***			13.04 ***		

* *p* < 0.05. *** *p* < 0.001. CI represents Confidence Interval.

**Table 3 behavsci-15-00589-t003:** The bootstrapping analysis of the mediating effects.

	Effect	*SE*	Boot 95%CI	Proportion
Total effect	0.33	0.04	0.14, 0.31	
Direct effect	0.27	0.05	0.18, 0.36	82.82%
Indirect effect	0.06	0.02	0.02, 0.10	18.18%

**Table 4 behavsci-15-00589-t004:** Testing the moderated mediation effect of rejection sensitivity on PIU.

Predictors	Model 2 (Loneliness)			Model 3 (PIU)		
	*b*	*t*	Boot 95%CI	*b*	*t*	Boot 95%CI
Gender	0.06	0.71	−0.11, 0.23	0.02	0.32	−0.13, 0.18
Age	−0.03	−0.63	−0.11, 0.06	0.02	0.46	−0.06, 0.10
Family location	0.11	1.38	−0.05, 0.27	0.11	1.52	−0.03,0.25
Grade	0.01	0.21	−0.10, 0.12	−0.06	−1.22	−0.16, 0.04
Rejection sensitivity	0.33 ***	8.38	0.25, 0.41	0.20 ***	5.02	0.12, 0.27
Self-control				−0.44 ***	−12.01	−0.51, −0.36
Loneliness				0.09 *	1.97	0.00, 0.17
Self-control × Loneliness				0.08 *	2.35	0.01, 0.16
*R* ^2^	0.15			0.38		
*F*	15.77 ***			33.35 ***		

Note: × represents the interaction item of Rejection sensitivity and Self-control. * *p* < 0.05, *** *p* < 0.001.

**Table 5 behavsci-15-00589-t005:** The conditional indirect effects.

Mediator	Self-Control	Effect	*SE*	Boot 95%CI
Loneliness	*M − SD*	0.01	0.05	0.01, 0.10
	*M*	0.08	0.04	0.01, 0.17
	*M + SD*	0.16	0.05	0.05,0.28

## Data Availability

The datasets utilized and/or analyzed during the present study are accessible upon reasonable request from the corresponding author.

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
