# Peer review of "Rejection Sensitivity and Problematic Internet Use Among Medical Students: A Moderated Mediation Model Involving Loneliness and Self-Control"

_behavsci, 2025, doi:10.3390/bs15050589_

Round 1
Reviewer 1 Report
Comments and Suggestions for Authors
The study investigates the relationship between rejection sensitivity and problematic internet use (PIU) among medical students, with loneliness as a mediator and self-control as a moderator.
Recommended Revision
Introduction:
- The discussion about rejection sensitivity leading to loneliness and internet addiction is repeated in multiple paragraphs.
- The introduction should clarify why medical students are particularly vulnerable to PIU compared to other student populations.
- The section jumps between discussing PIU prevalence, negative effects, psychological mechanisms, and hypotheses without clear transitions.
- The introduction does not explicitly state what past studies lacked.
Method
- The method section does not specify clear inclusion/exclusion criteria for the participants.
- Inconsistencies in Sample Characteristics, the reported number of students in different years does not match the total sample size. “The sample comprised 217 male students and 218 female students, with 136 first-year students, 167 second-year students, 59 third-year students, 55 fourth-year students, and 21 fifth-year students.” Total of all years = 136 + 167 + 59 + 55 + 21 = 438, but sample size is 435. There is a discrepancy of 3 students.
- In the Rejection Sensitivity Scale It does not state how the score is calculated (sum, mean, factor scoring?). The reliability (Cronbach’s alpha) is given as 0.82 but without reference to past validation studies.
- In the Problematic Internet Use (PIU) Scale the scoring criteria are not provided (e.g., cutoff for mild, moderate, severe addiction).
- In the Self-Control Scale Needs More Detail, the measurement description is vague. It does not mention scoring direction (higher = better self-control?). Two subscales are mentioned ("self-discipline" and "impulse control") but not explained.
Results
- The study does not explain why only age and gender were controlled, while other potential confounders (e.g., socioeconomic status, academic year) were ignored.
- The correlation table is missing effect size interpretations (e.g., weak, moderate, strong).
- Negative values are not clearly explained in relation to self-control.
- In Mediation Analysis, the effect size (ab = 0.06) is small, but no justification is provided. No total effect is reported (needed for full mediation interpretation). Confidence interval (CI) interpretation is missing.
- In Moderation Analysis the effect is described as "positive," but no explanation of the practical meaning is given. The p-value of 0.001 does not match the t-statistic (t = 2.26) (likely an inconsistency).
- Missing Effect Size for Moderation, the R² change due to the interaction term is not reported, which is crucial to assess the additional variance explained by moderation.
In the other sections
- The discussion section repeats findings already stated in the results section without further analysis.
- The Cognitive-Behavioral Model and Affect Regulation Model are mentioned in the introduction, but not well-integrated into the discussion.
- The discussion mentions prior studies but does not compare how the current findings support, contradict, or extend previous work.
- The implications are too general and do not offer concrete intervention strategies.
Comments on the Quality of English Language
The article is available as a preprint on ResearchSquare with the DOI: https://doi.org/10.21203/rs.3.rs-3958231/v1
With a detected plagiarism score of 68%.
Author Response
Please see the attachment, thanks very much.

Reviewer 2 Report
Comments and Suggestions for Authors
This manuscript is well-written and advances the literature by deepening scholars' understanding of theoretical frameworks and shedding light on the developmental pathway that leads to problematic internet use, which is a growing topic of concern. Although other studies have also looked into the Cognitive Behavioral Model and added to the understanding of the framework, these authors look into rejection sensitivity, an underexplored variable and its influence. They utilize validated scales and conduct a moderated mediation analysis that shows the positive influence of rejection on problematic use in medical students. They also address limitations including the cross-sectional nature of the study and provide reasonable suggestions for future research to explore.
Author Response
This manuscript is well-written and advances the literature by deepening scholars' understanding of theoretical frameworks and shedding light on the developmental pathway that leads to problematic internet use, which is a growing topic of concern. Although other studies have also looked into the Cognitive Behavioral Model and added to the understanding of the framework, these authors look into rejection sensitivity, an underexplored variable and its influence. They utilize validated scales and conduct a moderated mediation analysis that shows the positive influence of rejection on problematic use in medical students. They also address limitations including the cross-sectional nature of the study and provide reasonable suggestions for future research to explore.
Response: Thanks for your kindly comments. We made some revisions in revised manuscript.
Reviewer 3 Report
Comments and Suggestions for Authors
The article tackles a very important issue and the chosen method makes sense and applied systematically producing interesting evidence about the connections between the rejection sensitivity, loneliness and internet use among the medical students in Fuzhou City. It appears that the chosen sample with its high reply rate was very motivated and earnest to analyze their personal situation. For me this articles raises many issues how these results may be related to the particular situation of the chosen sample. In other words, it would be interesting to compare what kind of results would be attained with different kind of students in different societies and how important factors rejection sensitivity and loneliness would turn out in other studies. Anyway, this research tells a lot about the medical students in Fuzhou City and their ways of coping with rejection and loneliness. I would not go as far as to suggest that there is a universal connection between medical students and the behavioral pattens that were identified in this study. Personally I would be a bit more careful when drawing conclusions in the implications & conclusions parts of this article. However, the limitations of the study have also been clearly stated.
The specific comments:
In my review (see above) I pointed out that this article’s main contribution is to provide good analysis of the ways that medical students in Fuzhou City cope with rejection and loneliness. After that it would be desirable to try to place these results better in the context of studies that have been conducted elsewhere and then to provide ideas/ explanations how the coping mechanisms of medical students in Fuzhou City differ or resemble those in other locations in China and elsewhere. I would suggest rephrasing the parts related to conclusions & implications as well as providing more specific analysis of the limitations of this study as it exists now. I truly believe that this study has lot of significance and could well inspire research elsewhere. However, I frankly don’t know enough about the medical students in Fuzhou City to make any generalizations about their behavior and to compare them with medical students elsewhere.
Author Response
The article tackles a very important issue and the chosen method makes sense and applied systematically producing interesting evidence about the connections between the rejection sensitivity, loneliness and internet use among the medical students in Fuzhou City. It appears that the chosen sample with its high reply rate was very motivated and earnest to analyze their personal situation. For me this articles raises many issues how these results may be related to the particular situation of the chosen sample. In other words, it would be interesting to compare what kind of results would be attained with different kind of students in different societies and how important factors rejection sensitivity and loneliness would turn out in other studies. Anyway, this research tells a lot about the medical students in Fuzhou City and their ways of coping with rejection and loneliness. I would not go as far as to suggest that there is a universal connection between medical students and the behavioral pattens that were identified in this study. Personally I would be a bit more careful when drawing conclusions in the implications & conclusions parts of this article. However, the limitations of the study have also been clearly stated.
The specific comments:
In my review (see above) I pointed out that this article’s main contribution is to provide good analysis of the ways that medical students in Fuzhou City cope with rejection and loneliness. After that it would be desirable to try to place these results better in the context of studies that have been conducted elsewhere and then to provide ideas/ explanations how the coping mechanisms of medical students in Fuzhou City differ or resemble those in other locations in China and elsewhere. I would suggest rephrasing the parts related to conclusions & implications as well as providing more specific analysis of the limitations of this study as it exists now. I truly believe that this study has lot of significance and could well inspire research elsewhere. However, I frankly don’t know enough about the medical students in Fuzhou City to make any generalizations about their behavior and to compare them with medical students elsewhere.
Response: Thank you for pointing this out. We agree with this comment. We have revised the conclusions and implications, we provided more specific analysis of the limitations of the study. The revisions were highlighted in blue font in revised manuscript, and was also shown as follows:
Implication
Our findings have important practical implications. First, rejection sensitivity positively correlated with PIU among medical students, educational administrators need to pay attention to medical students' sensitivity in interpersonal relationships and its negative impact. Conducting relevant social skills courses and diverse group socialization activities, as well as creating an inclusive and tolerant microsystem environment, can increase social interactions among medical students and reduce their rejection sensitivity, preventing them from compensating through excessive use of the Internet. Second, our research findings provide insights for practitioners into the mediating role of loneliness in the influence of rejection sensitivity on PIU. The risk of PIU can be reduced by reducing rejection sensitivity and loneliness in medical students through counseling or social-emotional competence training, or by developing rich and varied community activities. Third, self-control moderates the relationship between loneliness and PIU. Therefore, educators need to recognize the role of self-control in preventing PIU in medical students and implement self-control-enhancing related curricula to improve medical students' self-control.
Limitation and Future Study
This study has some limitations. Firstly, due to its cross-sectional design, it is challenging to establish definitive causal relationships between variables, and the potential impact of social desirability bias on self-reporting by medical students should be acknowledged. Future research could employ longitudinal designs and experimental studies to better collect data and further examine the impact mechanisms of rejection sensitivity on internet addiction. Secondly, the participants in this study were limited to one city in mainland China, caution should be exercised when generalizing these results to other regions. Future research could expand the sampling area and size to explore whether the mediation model established in this study is universal, and medical students are not compared with other types of students. Meanwhile, medical students can be compared with other types of students to emphasize the unique characteristics of medical students. Thirdly, in current study, we assessed the PIU of medical students mainly in terms of duration, overdependence, degree of impact of Internet access on academics, and annoyance of not being able to access the Internet, without classifying specific types of Internet use. In future studies, we could categorize Internet use into specific subcategories, such as video games and chatting, and explore whether the effects of loneliness and self-control on the association between rejection sensitivity and PIU depend on particular type of internet activity. Lastly, rejection sensitivity includes the emotional component and the cognitive component.However, this study only investigated the effect of overall rejection sensitivity on PIU, and it is possible that these two dimensions have different effects on PIU, and future studies could further investigate the effect of specific dimensions of rejection sensitivity on PIU.
Conclusions
Based on the cognitive-behavioral model and the affect regulation model of vulnerability, The study advances our empirical understanding as to how and when medical students' rejection sensitivity influences their PIU. Our founding suggests that rejection sensitivity predicts PIU, and loneliness operates as a potential mechanism mediating this relationship. Furthermore, moderated mediation findings indicate that self-control acts as a moderator in the association between loneliness and PIU among medical students.
The change can be found in the revised manuscript (page number 8-9, paragraph 8:4; 9:2-3, and line 301-347. )
Reviewer 4 Report
Comments and Suggestions for Authors
I would like to thank the editor of this prestigious journal for the opportunity to evaluate this study and I would also like to congratulate the researchers for the effort made and the results obtained.
First of all, I find the study interesting and a good starting point for other researchers in designing future implications of internet/social media consumption in academia.
The research is well thought out, all parts of the manuscript are adequately addressed and show the thoroughness of the researchers in all areas.
I consider that the study can be improved in some aspects, which I summarise below:
Although it is indicated as a limitation, it talks about internet use in a very general way, which includes any use, when in fact the study seems to be more focused on the use of digital platforms or social networks. We can do many things on the internet, internet use is very broad, I think that what can most cause us to change our behaviour is the use of social networks, this is perceived in the paper but I think it should be emphasised, and above all indicate in the limitations of the study that there may be social networks or habits of use of social networks that influence more than others.
Talking about the internet and not about social networks can be a very important limitation to the study given the limitation that the country where the study is carried out has with respect to the rest of the world, as China has its own social networks.
I miss the fact that no questions have been asked about which social networks are used, how they are used (if they are content creators, opinion makers or just readers), topics of interest, I think that all these variables have a great influence on the hypotheses put forward in the study, it is not the same to use the internet to play games, watch films, expand knowledge or... social networks.
Another aspect that I find curious is that medical students are not compared with other types of students, I understand that there will not be similar studies in other disciplines of study but, what variables to study can influence differently to medical students than to others? If we can't just leave it to undergraduates. I think that more work needs to be done in this area.
I congratulate the researchers for the excellent research they have done and I am grateful for the opportunity to have been able to evaluate this study.
Author Response
I would like to thank the editor of this prestigious journal for the opportunity to evaluate this study and I would also like to congratulate the researchers for the effort made and the results obtained.
First of all, I find the study interesting and a good starting point for other researchers in designing future implications of internet/social media consumption in academia.
The research is well thought out, all parts of the manuscript are adequately addressed and show the thoroughness of the researchers in all areas.
I consider that the study can be improved in some aspects, which I summarise below:
Although it is indicated as a limitation, it talks about internet use in a very general way, which includes any use, when in fact the study seems to be more focused on the use of digital platforms or social networks. We can do many things on the internet, internet use is very broad, I think that what can most cause us to change our behaviour is the use of social networks, this is perceived in the paper but I think it should be emphasised, and above all indicate in the limitations of the study that there may be social networks or habits of use of social networks that influence more than others.
Talking about the internet and not about social networks can be a very important limitation to the study given the limitation that the country where the study is carried out has with respect to the rest of the world, as China has its own social networks.
I miss the fact that no questions have been asked about which social networks are used, how they are used (if they are content creators, opinion makers or just readers), topics of interest, I think that all these variables have a great influence on the hypotheses put forward in the study, it is not the same to use the internet to play games, watch films, expand knowledge or... social networks.
Another aspect that I find curious is that medical students are not compared with other types of students, I understand that there will not be similar studies in other disciplines of study but, what variables to study can influence differently to medical students than to others? If we can't just leave it to undergraduates. I think that more work needs to be done in this area.
I congratulate the researchers for the excellent research they have done and I am grateful for the opportunity to have been able to evaluate this study.
Response: Thank you for pointing this out. We agree with this comment. We sorry that we didn’t investigated the detail social networks. In the study, we assessed the PIU of medical students mainly in terms of duration, overdependence, degree of impact of Internet access on academics, and annoyance of not being able to access the Internet, without classifying specific types of Internet use. We use the Internet Addiction Test (YIAT) developed by Young which contain 20 items such as “How often do others in your life complain to you about the amount of time you spend on-line?. How often do your grades or school work suffers because of the amount of time you spend on-line?. How often does your job performance or productivity suffer because of the Internet?. How often do you fear that life without the Internet would be boring, empty, and joyless?.How often do you snap, yell, or act annoyed if someone bothers you while you are on-line?.How often do you lose sleep due to late-night log-ins?.How often do you choose to spend more time on-line over going out with others?” .Social networks may have a significant connection with the rejection sensitivity and loneliness of medical students. In future studies, we could categorize Internet use into specific subcategories, such as video games and chatting, and explore whether the effects of loneliness and self-control on the association between rejection sensitivity and PIU depend on particular type of internet activity. We also put it into limitation in the revised manuscript. In addition, we didn’t compare with other types of students which is a limitation, we also put it into the revised manuscript, and we hope that these elements can be taken into account in the relevant research in the future. Thanks very much for your suggestions.
The revision was shown as follows, and it also has been highlighted in blue font in revised manuscript.
“Limitation and Future Study
This study has some limitations. Firstly, due to its cross-sectional design, it is challenging to establish definitive causal relationships between variables, and the potential impact of social desirability bias on self-reporting by medical students should be acknowledged. Future research could employ longitudinal designs and experimental studies to better collect data and further examine the impact mechanisms of rejection sensitivity on internet addiction. Secondly, the participants in this study were limited to one city in mainland China, caution should be exercised when generalizing these results to other regions. Future research could expand the sampling area and size to explore whether the mediation model established in this study is universal, and medical students are not compared with other types of students. Meanwhile, medical students can be compared with other types of students to emphasize the unique characteristics of medical students. Thirdly, in current study, we assessed the PIU of medical students mainly in terms of duration, overdependence, degree of impact of Internet access on academics, and annoyance of not being able to access the Internet, without classifying specific types of Internet use. In future studies, we could categorize Internet use into specific subcategories, such as video games and chatting, and explore whether the effects of loneliness and self-control on the association between rejection sensitivity and PIU depend on particular type of internet activity. Lastly, rejection sensitivity includes the emotional component and the cognitive component.However, this study only investigated the effect of overall rejection sensitivity on PIU, and it is possible that these two dimensions have different effects on PIU, and future studies could further investigate the effect of specific dimensions of rejection sensitivity on PIU."
The change can be found in the revised manuscript (page number 9, paragraph 2; 9:2-3, and line 317-339 )
Reviewer 5 Report
Comments and Suggestions for Authors
Dear Authors,
Congratulations on your interesting article. I personally have no objections to the content of the paper, but I have noticed some writing errors, such as spacing issues, parentheses, and similar formatting inconsistencies. These should be corrected to improve the overall clarity and presentation of the manuscript.
Best regards
Author Response
Congratulations on your interesting article. I personally have no objections to the content of the paper, but I have noticed some writing errors, such as spacing issues, parentheses, and similar formatting inconsistencies. These should be corrected to improve the overall clarity and presentation of the manuscript.
Response:Thank you for pointing this out. We agree with this comment. We corrected the writing errors in revised manuscript including spacing issues and parentheses. We are sorry for the mistakes. Thanks for your suggestions.
Best regards
Round 2
Reviewer 1 Report
Comments and Suggestions for Authors
Nothing